# Applications for Deep Learning in Epilepsy Genetic Research

**DOI:** 10.3390/ijms241914645

**Published:** 2023-09-27

**Authors:** Robert Zeibich, Patrick Kwan, Terence J. O’Brien, Piero Perucca, Zongyuan Ge, Alison Anderson

**Affiliations:** 1Department of Neuroscience, Central Clinical School, Monash University, Melbourne, VIC 3800, Australia; robert.zeibich@monash.edu (R.Z.); patrick.kwan@monash.edu (P.K.); terence.obrien@monash.edu (T.J.O.); piero.perucca@unimelb.edu.au (P.P.); 2Department of Neurology, Alfred Health, Melbourne, VIC 3004, Australia; 3Department of Neurology, The Royal Melbourne Hospital, The University of Melbourne, Parkville, VIC 3052, Australia; 4Department of Medicine, The Royal Melbourne Hospital, The University of Melbourne, Parkville, VIC 3052, Australia; 5Epilepsy Research Centre, Department of Medicine, Austin Health, The University of Melbourne, Melbourne, VIC 3084, Australia; 6Bladin-Berkovic Comprehensive Epilepsy Program, Department of Neurology, Austin Health, The University of Melbourne, Melbourne, VIC 3084, Australia; 7Faculty of Engineering, Monash University, Melbourne, VIC 3800, Australia; zongyuan.ge@monash.edu; 8Monash-Airdoc Research, Monash University, Melbourne, VIC 3800, Australia

**Keywords:** machine learning, deep learning, genetic epilepsy, non-protein-coding, omics data integration

## Abstract

Epilepsy is a group of brain disorders characterised by an enduring predisposition to generate unprovoked seizures. Fuelled by advances in sequencing technologies and computational approaches, more than 900 genes have now been implicated in epilepsy. The development and optimisation of tools and methods for analysing the vast quantity of genomic data is a rapidly evolving area of research. Deep learning (DL) is a subset of machine learning (ML) that brings opportunity for novel investigative strategies that can be harnessed to gain new insights into the genomic risk of people with epilepsy. DL is being harnessed to address limitations in accuracy of long-read sequencing technologies, which improve on short-read methods. Tools that predict the functional consequence of genetic variation can represent breaking ground in addressing critical knowledge gaps, while methods that integrate independent but complimentary data enhance the predictive power of genetic data. We provide an overview of these DL tools and discuss how they may be applied to the analysis of genetic data for epilepsy research.

## 1. Introduction

Affecting approximately one percent of the world’s population, epilepsy is one of the most common neurological disorders. Rather than being a single homogenous condition, epilepsy encompasses a group of heterogeneous brain diseases characterised by an enduring predisposition to generate epileptic seizures; hence, the term ‘epilepsies’ is more appropriate [1,2,3,4]. The epilepsies are classified into one of four main types according to the associated seizure types: focal epilepsies, characterised by seizures originating within brain networks limited to one hemisphere (focal seizures); generalized epilepsies, where seizures involve both hemispheres (generalised seizures); combined focal and generalized epilepsies; epilepsies of unknown type for which there is insufficient information to classify them (seizures of unknown onset) [5,6]. Within these broad groups, there are specific epilepsy syndromes that are often supported by specific aetiological findings (structural, genetic, metabolic, immune, and infectious [7].

There are substantial genetic contributions to the aetiology of many of the epilepsies, particularly in the developmental and epileptic encephalopathies (DEEs), a group of rare severe epilepsies. In this epilepsy group, approximately 50% of cases can now be attributed to either inherited or de novo variants (not inherited from parents) [8,9]. Major progress has been made in elucidating a role for genetic variance in focal epilepsies, which account for 60% of all epilepsies and which had historically been considered as largely acquired disorders [10]. Between 10 and 15% of individuals with focal epilepsy and a family history of epilepsy are attributable to germline pathogenic variants [11,12]; the ‘hit rate’ increases to 30–40% in those with brain malformations, which are largely accounted for by genetic variants that occur after birth (somatic mutations) [13]. The term genetic generalised epilepsies (GGEs) is used to describe patients with generalised seizure types that are presumed to have a genetic aetiology [14]. GGEs account for 15–20% of all epilepsies [14,15]. Within the GEE are two common ‘absence epilepsies’, childhood absence epilepsy and juvenile absence epilepsy, which together with the syndromes of juvenile myoclonic epilepsy and generalised tonic-clonic seizures alone form the idiopathic generalised epilepsies (IGEs) [14]. The main contribution to this epilepsy group appears to be from a large collection of common variants each conferring a small effect size [16].

To identify genes implicated in epilepsy (as well as in other disorders), single individuals, mother father and child trios, whole families, and large cohorts are oftentimes tested. The technologies used for testing include chromosomal microarrays (CMAs) for the detection of large structural variation in the DNA, single-nucleotide polymorphism (SNP) array data that capture common variation and sequencing or whole genome sequencing (WGS). The use of whole exome sequencing (WES) and WGS is increasingly feasible as the costs for short-read (150 base pairs) technology will continue to decrease [17]. Technologies that read longer lengths (5000–30,000 base pairs) generated by Pacific Biosciences’ (PacBio) single-molecule real-time (SMRT) sequencing and the Oxford Nanopore Technology (ONT) platform can improve on short-read sequencing for tasks such as detecting structural variation [18], but are often challenged by increased error rates [19]. The development of bioinformatic methods to resolve the error rates is important as these tools have attractive clinical utility given the ability to sequence in real time; a pipeline using the ONT PromethION found candidate disease variants within 8 h of blood draw [20] and could potentially be used to detect variants relevant to pharmacogenomics.

The processing of genomic data to yield useful clinical information can be broadly grouped into three steps: variant detection, variant annotation and interpretation, and classification or prediction tasks (see Figure 1 and Box 1).

Box 1Brief glossary of genetic terms. **Protein coding regions:** DNA regions, also known as exons, that contain the instructions for producing a protein (regions within genes). **Intronic regions:** DNA regions that are located within introns (non-coding regions within genes). **Intergenic regions:** DNA regions that are located between genes (non-coding regions between genes). **Variants:** Differences in DNA sequences compared to a reference genome. **Single-nucleotide variants (SNVs):** Genetic variation that occurs when a single nucleotide differs from the reference genome. **Insertions and deletions (Indels):** Genetic variations that involve the insertion or deletion of DNA sequences. **Structural variants (SVs):** Genetic variations that involve large-scale changes in the structure of the genome, including deletions, insertions, duplications, insertions, inversions, or translocations. **Copy number variants (CNVs):** Structural genetic variations that involve changes in the number of copies of a particular DNA region. **Rare variants:** Rare genetic variations that are generally considered to be present in less than 1% of the population. **Common variants:** Common genetic variations that are generally considered to be present in at least 1% or 5% of the population. **Mosaic variants:** Genetic variations that are not expressed in all germline, somatic, or both cells. **Candidate disease variants**: Genetic variations that are hypothesized to be associated with a particular trait or disease based on prior knowledge/evidence. **Variant interpretation:** The process of assessing the significance or pathogenicity (likely benign, benign, variant of uncertain significance, likely pathogenic, or pathogenic) and determining the potential functional consequences of genetic variations. **Variant browser:** Software tools or web-based interface that allows the visualization of genetic variations. **Polygenic risk scores (PRS):** Calculated score based on multiple common genetic variations, which estimates the risk of developing a particular trait or disease. **Risk allele:** Genetic variation within a gene that is associated with an increased risk of a specific trait or disease. **Gene panels:** Targeted sets of genes used to assess rare genetic variations of large effect size based on their suspected involvement in a particular trait or disease. **Whole genome**: Genome representing the complete set of genetic information (DNA sequence) of an organism. **Whole exome:** Subset of a genome containing only protein-coding regions (exons) of genes. **Single-nucleotide polymorphism (SNP) array data:** Data generated using SNP arrays to detect and analyse SNVs across a genome. **Chromosomal microarrays (CMAs):** Technique used to detect chromosomal abnormalities or copy number variations in the genome. **Short-read sequencing data:** Data generated through next-generation sequencing technologies that produce DNA or RNA fragments ranging from a few dozen to a few hundred base pairs in length. **Long-read sequencing data:** Data generated through sequencing technologies that produce DNA or RNA fragments ranging from several thousand to tens of thousands of base pairs in length. **Gene expression data:** Data that quantifies the abundance or activity of genes in a specific sample or tissue. **Genotype data:** Data representing genetic information at specific genomic positions, typically focusing on SNVs or small insertions and deletions. **Protein–protein and gene co-expression networks:** Networks that represent the interactions or co-expression patterns between proteins or genes, respectively. **Genome-wide association study (GWAS):** Study designed to identify common genetic variations raising the risk for traits, diseases, or phenotypes on a genome-wide scale. Typically involves genotyping or sequencing on a large scale and comparing common genetic variations between cases and controls to identify statistically significant associations.

The development of methods and tools for processing and interpreting sequencing data is an active area of research and as with many other fields, machine learning is playing a transformational role. Machine learning (ML) is a type of artificial intelligence that uses statistical algorithms to determine an output from a given input. Deep learning (DL) is a subfield of ML that was inspired by the complexity of neuronal networks in the brain and uses artificial neural network architectures to perform ML tasks [24]. DL models have multiple layers of nonlinear processing units that progressively learn representations of the data needed for detection or classification [25]. In epilepsy, DL has been successfully applied for the study of magnetic resonance imaging (MRI) [26], diffusion kurtosis imaging (DKIs) [27], and electroencephalogram (EEG) [28]. Here, we review DL methods and tools (see Box 2) with the capacity to address challenges or enhance genomic research tasks as outlined in Figure 1.

Box 2Brief glossary of DL-related terms. **Supervised Machine Learning:** Models are trained on labelled samples to subsequently classify the labels of unseen samples. Commonly used algorithms are Decision Trees, Linear Regression, Logistic Regression, and Support Vector Machines (SVM). **Unsupervised Machine Learning:** Models that identify clusters in data from un-labelled data. Commonly used algorithms are Principal Component Analysis (PCA) and K-means clustering. Semi-supervised Machine Learning models are trained on labelled and un-labelled samples. **Deep Learning (DL):** DL uses artificial neural network architectures to perform ML tasks. The most basic neural network encompasses three layers with nodes or neurons: an input, hidden, and output layer. Once the number of hidden layers is increased, the network increases in depth and is then considered to be a deep neural network. **Convolutional Neural Networks (CNN):** A type of network used for spatial data, images, video, or sequences. The CNN extracts local patterns (e.g., edges or sequence patterns) by applying a filtering mechanism whereby the output from the previous layer is passed to the next layer. **Recurrent Neural Networks (RNN):** Networks that can process sequential or temporal data and are, therefore, predominantly used for speech recognition, natural language processing (NLP) or processing time series data. **Transfer learning:** Methods that improve the learning process by transferring previously learnt knowledge (e.g., parameters, weights, or output data) from an existing ML model or external information (source) to a new ML model. **Data augmentation:** Methods that improve the performance of DL by adding newly synthetically created or slightly modified samples to increase the number of training samples. **Representation learning:** Algorithms can learn representations (or features) of raw data that improve the performance of a model or be used as input to a model. Deep learning allows multiple transformations, yielding more abstract and potentially more useful representations. **Overfitting and Underfitting:** Terms applied to a model that performs poorly when tested on new data. The model might perform well on test data but not on new unseen data (overfitting), or make incorrect predictions with new data (underfitting). **Dimensionality Reduction:** Methods applied before modelling to reduce the number of features while preserving the relationships between features in data. **Feature Learning or Feature Engineering:** Techniques that map the data into a more consolidated or lower dimension while trying to preserve the input information. **Generative Adversarial Networks (GANs):** Networks that encompass two neural networks, a discriminator and a generator network, which are both trained together. The generator produces data samples, and the discriminator differentiates whether a given sample is real or generated by the generator. **Autoencoder:** An unsupervised learning architecture that encompasses two neural networks, a feature extractor (encoder) and a feature validator (decoder) that work together to generate a representation of the data with reduced dimensionality.

## 2. Variant Calling

The Genome Analysis Toolkit (GATK) is a software pipeline developed at the Broad Institute [29] for processing short-read next-generation sequencing data. It has become the industry ‘gold standard’ for identifying single-nucleotide variants (SNVs) and short insertions and deletions (indels) from short-read germline DNA sequence data. In recent years, several new DL tools for variant calling, which involves comparing the sequence of an individual against a reference genome to identify differences within the genome, have been introduced. Short-read callers include DeepVariant [30] and HELLO (Hybrid and stand-alone Estimation of small genomic variants) [31]. DeepVariant [30] generates an image by mapping the raw read data to a 2D data matrix and then uses the images to call variants based on predictions for candidate alternative alleles at each site. In contrast, the HELLO algorithm uses a deep neural network and customized variant inference functions to make predictions for each candidate allele by analysing the reads supporting each allele. The authors of HELLO claim that their method outperforms DeepVariant, in terms of a reduced number of insertion and deletion errors and accuracy [31]. Clair [32] is a recently introduced deep neural network caller that was specifically designed to reduce the error rate inherent in long-read sequencing data but performs less well, relative to DeepVariant, on short-read data. The authors claim that Clair outperforms their earlier DL tool called Clairvoyant, and two other long-read callers introduced in 2019 named Longshot [33] and Medaka [32]. The ONT MiniON platform uses a method known as ReadUntil which allows unwanted sequences to be depleted during the sequence run. UNCALLED is a variant caller that exploits the ReadUntil method. When tested using a panel of human cancer genes, it was found to have high precision in the detection of SNVs, indels, structural variations and DNA methylation [34].

As new tools are introduced, improved performance is typically demonstrated by comparison against earlier published methods and using standard genomes for benchmarking made available through, for instance, the gene in a bottle consortium (https://www.nist.gov/programs-projects/genome-bottle; accessed on 20 September 2023). Evidence of how these tools perform in real-world research and clinical settings is limited, and it will take time to generate sufficient data for robust evaluation. A comparison between the GATK pipeline and DeepVariant for the detection of pathogenic variants from exome data from independent cohorts of cancer patients found that DeepVariant had higher sensitivity and specificity [35]. In epilepsy, the adoption of WGS is gaining momentum. A systematic review and meta-analysis of genetic testing options in epilepsy found WGS to have the highest diagnostic yield (48%) [9]. Costain, Cordeiro, et al. [36] found WGS gave the highest yield for childhood epilepsy. Ostrander, Butterfield, et al. [37] evaluated the effectiveness of WGS for clinical diagnosis and gene discovery in early infantile epileptic encephalopathy and concluded that, in comparison to standard approaches involving multiple genetic tests, WGS saves time and costs while also allowing for the evaluation of non-coding and CNVs. The American Society of Genetic Counsellors also recommends genetic testing, preferentially exome or genome sequencing or multi-gene panel testing, for individuals with unexplained epilepsy, irrespective of age [38]. The GREP (Genomic sequencing for Refractory EPilepsy) study [39], a randomised controlled trial, aims to assess the utility and cost-effectiveness of WGS for refractory epilepsy in children and adults will provide a valuable ‘ground truth’ of patient outcomes for future performance testing. The EMPOWER-1: A Multi-site Clinical Cohort Research Study to Reduce Health Inequality in the United Kingdom is using WGS to identify candidate genetic variants that may underpin observed disparities in treatment failure for 19 disease areas, including epilepsy [40].

## 3. Variant Annotation

Variant annotation tools provide information about the likely functional consequence of genetic variants. This information is then used to evaluate, interpret, and classify variants in accordance with the American College of Medical Genetics (ACMG) guidelines (see Figure 2) [41].

Tools commonly used for annotation include the Ensembl Variant Effect Predictor (VEP) [42], ANNOVAR (ANNOtate VARiation) [43], and snpEff [44]. The annotation information generated includes, for example, scores generated by the PolyPhen-2 (Polymorphism Phenotyping v2) tool that indicate the possible impact of an amino acid substitution on the structure and function of a human protein [42,43,44]. Some tools, such as the Franklin by genoox web-based tool (https://franklin.genoox.com/clinical-db/home; accessed on 20 September 2023) introduced in 2019, and the MobiDetails online tool for DNA variant interpretation [45] introduced in 2020, automatically classify variants based on the American College of Medical Genetics (ACMG) guidelines [41]. MobiDetails is freely available for academic use (https://mobidetails.iurc.montp.inserm.fr/MD/; accessed on 20 September 2023) and is designed to work on mobile telephones or tablets by medical staff. This ‘real time’ interpretation of genetic variation will be necessary to keep pace with the evolution of ‘clinic-friendly’ real-time DNA sequencing tools.

### 3.1. Protein-Coding Variants

Numerous tools have been developed to predict whether amino acid substitutions result in disease. Output from these tools is typically used as ‘supporting’ evidence for variant interpretation and classification [46]. The ACMG recommends that this evidence be counted only if all the in-silico tools used to predict are concordant in their findings (e.g., all predict deleterious or all predict benign effect). This is problematic as tools vary in accuracy with more recent algorithms have higher predictive power over older poorer poorer-performing algorithms. Also, concordant in silico predictions have been found to be opposite to the evidence provided by other sources leading to an error in variant interpretation, that has been termed ‘false concordance’ [47]. Metapredictors that incorporate older algorithms as features (e.g., REVEL) may be a better alternative, but still not ideal given the inefficiencies of older algorithms and duplication of information [47].

DL and innovative modelling architectures hold promise for improving prediction. For example, the graphical missense variant pathogenicity predictor (gMPV,) is a tool for predicting the impact of missense variants. It uses a novel graph attention neuronal network that pools information on the variant with information on the local protein context (see Figure 3) [48].

### 3.2. Intronic and Intergenic Regions

Annotation information remains limited for variants that fall within intronic and intergenic DNA regions, yet these regions contain sequences with critically important roles such as splicing sites, binding sites for transcription factors, and CpG islands where DNA methylation occurs. Methylation within enhancer regions can have a similar function to methylation within gene promoter regions but needs yet to be fully understood [49,50]. Since DL tools are helping to address this knowledge gap, we provide below some examples of how DL tools predict the impact on these mechanisms, controlling how genes are transcribed and turned on or off.

#### 3.2.1. Transcription Factor Binding

Transcription factors (TFs) are proteins that bind to short DNA sequences (usually 6–12 bases) and act as ‘master regulators’ of cell type-specific gene expression [51]. Polymorphisms in TF binding sites comprise only 8% of the genome, yet represent 31% of all trait-associated polymorphisms [52]. Genetic variants within the binding sites can modify the strength of binding (binding affinity), prevent one or more TFs from binding at all, or permit aberrant TF binding [53]. A role for variant-disrupted TF binding has been evidenced in epilepsy, but remains poorly understood. In an animal model of epilepsy, a single long seizure was found to increase the levels of the TF neuron restrictive silencer factor (NRSF). The effect on genes regulated by NRSF was dependent on the binding affinity of this repressor at its target binding sites, with mid-range binding frequencies rendering genes sensitive to moderate fluctuations with deleterious effects [54]. The BCL11A TF was found to have the strongest association signal for the GGEs in the International League Against Epilepsy (ILAE) genome-wide association study (GAWS) [55]. Given that there are large numbers of TFs and their matched binding sites are widely dispersed throughout the genome, it is likely that this kind of genetic risk is highly heterogeneous and remains elusive. DeepBind [56] is a tool that can be used to investigate the impact of genetic variation on TF binding. This tool was trained using information from multiple sources including protein binding microarrays (PBMs), ribonucleic acid complete (RNAcomplete) data, chromatin immunoprecipitation sequencing (ChIP-seq) and powerful assays that determine the in vitro binding specificities of proteins (SELEX). This tool can be used to predict the binding affinity of 617 human-related TFs across the entire genome. The algorithm produces a score that represents the binding of a specific TF at specific DNA loci [53]. By comparing binding affinity scores generated using reference DNA and DNA-carrying variants at any specific binding site, this tool can be used to determine the impact of genomic variance on TF activity. This information can then be incorporated into annotation tools.

#### 3.2.2. DNA Methylation

DNA methylation involves the covalent addition of a methyl group at CpG sites and is the most-studied and best-understood of the epigenetic marks [57]. Variants that associate with changes in DNA methylation have been variously named methylation quantitative trait loci or meQTLs, mQTLs, or metQTLs [57]. Aberrant DNA methylation has been reported in patients with focal epilepsy [58], including temporal lobe epilepsy (TLE) [59] and, specifically, TLE with hippocampal sclerosis [60]. Animal models of epilepsy have implicated dysregulation of DNA methylation across different rat strains [61,62] and show that status epilepticus modifies the methylation status of the glutamate receptor Grin2b [63]. CpGenie [64] is a tool that was trained on methylation sequencing data to predict the methylation status at CpG sites in the DNA. It can evaluate the impact of a variant on methylation status by comparing predicted methylation at CpG sites flanking a variant, with predictions based on reference DNA at the same locus.

#### 3.2.3. Splicing

A large number of pathogenic variants affect how mRNA is spliced [65]. These variants can fall between splice-specific dinucleotides (GT and AG) or within other regions in the DNA where their presence introduces alternative splicing activity (cryptic splice variants) [66]. A role for splice variants in epilepsy is evidenced for genes associated with the DEEs and focal epilepsy including but not limited to STXBP1 [67], SCN1A [68], DNM1 [69], DEPDC5 [70], and GRIN2B [71]. Tools that predict variant impact have been available for a long time, but are constantly being improved (e.g., SpliceFinder introduced in 2019 [72] and SpliceViNCI in 2021 [73]). SpliceAI [66] is a DL predictor that is widely adopted and included in current annotation tools and data repositories including the Genome Aggregation Database (gnomAD) [74]. SpliceAI-visual is a recently introduced improved version that has been trained on additional data and overcomes limitations in the earlier version. It is freely available, compatible with the commonly used variant browsers (Integrative Genomics Viewer [IGV] and UCSC) and has been integrated into the MobiDetails variant interpretation tool [75]. This evolution makes it challenging for researchers to know which tool to use and highlight the importance of keeping variant curation pipelines up to date.

In summary, DL-based tools for predicting the functional impact of genetic variation are rapidly evolving and bring opportunities to break ground on annotation knowledge gaps and the ‘false concordance’ issue. This work is supported by the emergence of tool repositories that are made accessible through the open-source community platform, GitHub, which fosters community-based software development and helps to ensure robust and reusable code and methods. An example is the Kipoi repository [76], which provides tools that are specific to genomics. The tools are typically pre-trained for specific tasks and ready to use on new data. Bioinformaticians and software developers can also use them in a building block fashion to generate more comprehensive models or solve more complex tasks [76].

## 4. Multimodal Data Integration

In the DL literature, the terms ‘data fusion’, ‘information fusion’, and ‘multi-view’ and ‘multimodal’ learning are commonly used to describe models that integrate different data modalities [77,78]. The integration of genetic data, such as GWAS summary statistics, with data or information from complementary sources can lead to novel insights. Johnson et al. (2015) employed what they called a ‘systems genetic approach’ whereby they integrated network analysis of global gene expression in the hippocampi of TLE patients with GWAS data. Similarly, they integrated genetic risk data for psychiatric disease and behavioural traits with gene expression data in a rat model, to gain insight into neurodevelopmental outcomes following gestational exposure to the known teratogenic anti-seizure medication valproic acid [79].

de Jong, Cutcutache, et al. [23] explored how genetic and clinical data could be integrated into predicting response to brivaracetam, a new antiseizure medication. They generated several ‘genomic features’ from genotype data to use as input to their models including polygenic risk scores, a dichotomous variable for the presence of a specific gene variant, and burden scores for gene sets in which the genes were relevant to either epilepsy or the drug mode of action. They evaluated one DL and four ML integration strategies and found that the latter performed best. They considered that the poor performance of the DL method was likely due to the large number of features and the small number of samples [23].

Methodologies for multimodal modelling are rapidly evolving, although the application of these methods in medical research and clinical setting is still in its nascent stages. Li, Wang, et al. [80] provide a proof-of-concept for the integration of SNVs and brain image data to delineate schizophrenic patients and healthy controls. The model architecture was designed to detect complex nonlinear relationships between SNV and the image data. In epilepsy, Shen et al. [22], used a convolutional neural network (CNN) to predict post-stroke epilepsy. The model inputs include EEG signal data and the frequency of gene mutations in genes associated with stroke.

He and Xie [81] developed a method for the prediction of anti-cancer drug sensitivity named Cross-LEvel Information Transmission (CLEIT). As the name suggests the underlying method transmits information ‘learned’ from modelling one data type (e.g., gene expression) to a model of a second data type (e.g., genotypes) to improve the predictive capacity of the latter. This is an example of a strategy known as transfer learning. This method demonstrates that integration of genotype data with outputs learned from gene expression data can considerably improve performance over prediction with genotype data alone.

Integrating information and knowledge can also improve performance when modelling a single data type. Marini, Limongelli, et al. [82] used a method they describe as a ‘data fusion network’ to distinguish patients affected by epilepsy from controls. This method merged domain knowledge, gene, pathway, and protein–protein interaction data related to genes in an epilepsy gene panel. The integration of multiple data/knowledge sources can also be used to generate features for input to a model. This typically involves representation learning. For each data or information source, the algorithms discover the best representation of the raw data, then multiple representations are combined in a predictive model. This approach was utilised to develop a Multi-Graph Representation learning-based Ensemble Learning method (MGREL) for gene–disease association prediction [83].

## 5. Challenges and Potential Solutions

Insufficient data: DL algorithms work best when there are large numbers of samples available for training which is often not the case. Data augmentation [84] and transfer learning [85] are technical strategies that can be adopted to overcome insufficient datasets. Data augmentation involves the generation of new or slightly modified samples to increase the sample numbers for training the algorithms, which, in turn, improves the performance of the models [84]. Data augmentation has been successfully applied to epileptic seizure prediction from EEG data [21], and to delineate the seizure-onset zone in individuals with focal epilepsy [86]. See Habashi, Azab, et al. [87] for a full review. In genomics, Wei, Li, et al. [84] used data augmentation of genetic cancer data to demonstrate how this strategy improves cancer classification.

Transfer learning improves the learning process by transferring previously learnt knowledge (e.g., parameters, weights, or output data) from one model to another, as in the CLEIT method [81]. In general, by applying transfer learning, a new ML model can be trained in less time with fewer data but with increased predictive performance relevant to an ML model trained on available data in the target domain alone [76,88]. Si, Zhang, et al. [89] explored the utility of transfer learning combined with diffusion MRI to predict juvenile myoclonic epilepsy in a small sample of participants (15 with juvenile myoclonic epilepsy and 15 healthy controls). In genomics, Liu, Meng, et al. [90] propose a deep transfer learning model for the calculation of PRSs and Tan and Shen [91] show how transfer learning can be applied for in silico confirmation of rare CNVs identified from sequencing data.

High dimensionality: In genomics, the number of variables typically far exceeds the number of samples and too many features can lead to overfitting and poor model performance. This challenge is also commonly referred to as the curse of data dimensionality [92]. Conventional dimensionality reduction uses linear and non-linear transformation, including spectral [93] and kernel [94] methods to reduce the number of features into a lower dimension [95]. In genomics, important features can be lost when a linear transformation is applied [96]. The architecture of DL models supports non-linear dimensional reduction [95]. Representation learning is an important and increasingly applied solution. Representation learning can, for example, capture global features in networks representing protein–protein interactions and gene co-expression networks. Topological network information and a representation learning techniques have been used to identify genes associated with cerebral ischemic stroke [97], to prioritize candidate genes for complex diseases with gene networks and GWAS data [98], and to identify gene-phenotype relations from the biomedical literature [99].

Low interpretability: The condensed data representations generated by DL are challenging to interpret, leading them to be labelled as ‘black-box’ methods. The interpretability of a model can deteriorate as the complexity of the neural network (e.g., depth, width, integration of data modalities) is increased. Work on improving this issue is ongoing. Techniques that confer an ‘interpretable’ outcome have been demonstrated in a model that learned features from SNP and brain imaging data from a neurodevelopmental cohort [100]. This method identified specific SNPs and functional connectivity in the brain.

Further reading: For a broader summary of DL strategies, see Stahlschmidt, Ulfenborg and Synnergren [101], and the book “Deep Learning” by Bengio, Goodfellow and Courville [102].Stahlschmidt, Ulfenborg and Synnergren [101], and the book “Deep Learning” by Bengio, Goodfellow and Courville [102].

## 6. Conclusions

In this review, we have highlighted important roles for DL for variant calling, variant interpretation, and multimodal techniques for classification and prediction tasks relevant to epilepsy genetics research. The rapid developments in this field make it challenging for researchers and clinicians to keep abreast and to make the best tool and method selections. This is important as it has been shown that the choice of the variant caller or in silico tools used for variant annotation has an impact on diagnostic yield [47]. The ACMG guidelines criteria for in silico support for pathogenicity (PP3) and benign effects (BP4) are based on in silico predictor tool concordance but this becomes increasingly challenging as more predictors are tested. A recent evaluation of the impact of these criteria found that on the removal of the PP3 criterion, 14% of pathogenic and 24% of likely pathogenic variants were downgraded to likely pathogenic and VUS, respectively, while the removal of BP4 changed the classification of 64% of variants from benign to a variant of unknown significance (VUS) [103]. It is suggested that variant classification be considered a dynamic process, whereby there is benefit in the re-evaluation of previously classified variants as tools and methods evolve [104]. The rapid development of tools also creates a pressing requirement for benchmark datasets.

Repositories such as Kipoi aim to make it easier for researchers to adopt DL methods. Additional DL repositories relevant to genomic research include Selene [105], pyster [106], and Janguu [107]. Unlike commercial software, these tools require bioinformatics or coding expertise, and technical support may be limited, slowing down uptake by researchers and clinicians. Another caveat is that many of these tools have been trained on large publicly available data, in particular, ENCODE [108], but not all tissues or cell types are represented in these training data [109].

Multimodal modelling is an important next step in genomic research, with relevance to precision medicine. de Jong, Cutcutache, et al. [23] demonstrated how the inclusion of genetic information could improve predictive power and the CLEIT model by He and Xie [81] demonstrated how DL methods could better model the nuance and complexity in the relationships between features across data modalities. Publicly available repositories of information and genomic data will be increasingly important for enabling the training and evaluation of DL applications.

## Figures and Tables

**Figure 1 ijms-24-14645-f001:**
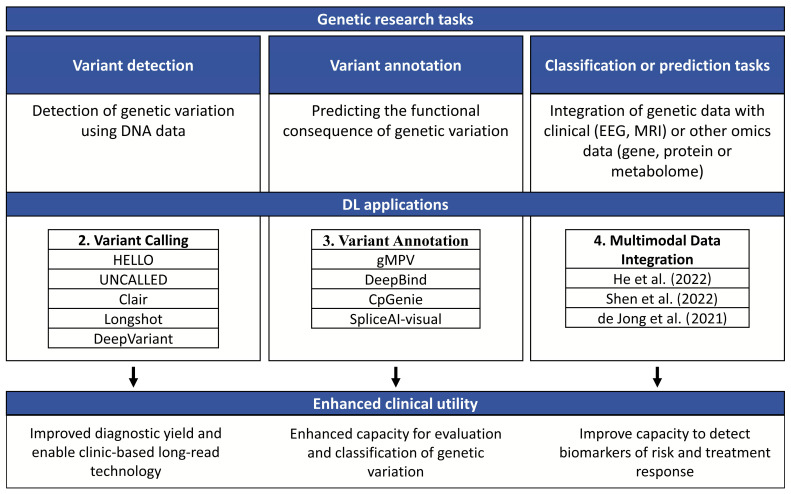
Routine clinical and genetic research tasks that can be enhanced by DL [21,22,23].

**Figure 2 ijms-24-14645-f002:**
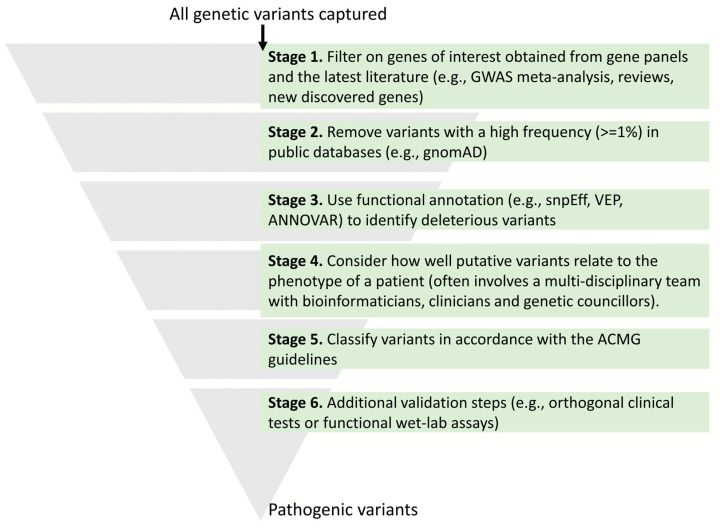
Overview of how genetic variants are filtered and screened.

**Figure 3 ijms-24-14645-f003:**
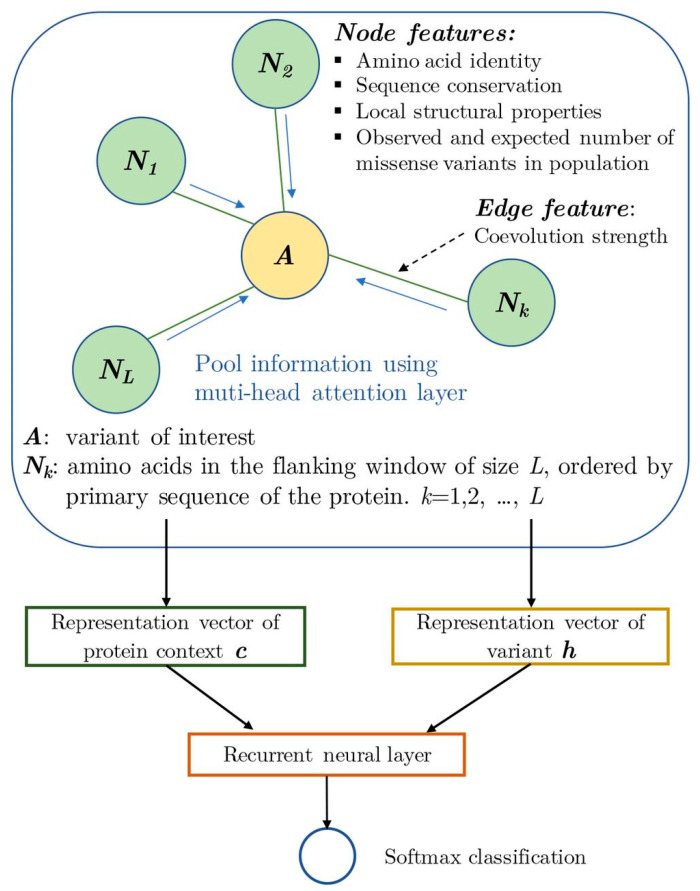
Overview of the graphical methodology that pools together feature information about a protein, represented as green nodes (*N*_1_, *N*_2_, *N_L_*, *N_k_*) in the network and genetic variation, represented as a yellow node (*A*). Edges, the lines that link nodes, can also contain information such as the strength of coevolution across species; copied and text adjusted from [48] licensed under Creative Commons BY 4.0 (CC BY NC ND 4.0). Full terms at https://creativecommons.org/licenses/by-nc-nd/4.0/.

## Data Availability

No new data were created or analyzed in this study. Data sharing is not applicable to this article.

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
