# Peer review of "Applications for Deep Learning in Epilepsy Genetic Research"

_ijms, 2023, doi:10.3390/ijms241914645_

Round 1
Reviewer 1 Report
Zeibich et al. present a very accurate and well-structured paper on the rapidly growing field of epilepsy genetics research.
The authors provide a very detailed introduction with accompanying comprehensive explanations of terms in 2 boxes.
In the main part, variant calling, variant annotation and multimodal data integration are discussed in 3 sections, and the challenges and potential solutions are discussed in the last section.
Mahor points:
1) Throughout, the target audience remains somewhat unclear. If this overview is intended for geneticists, a glossary is completely superfluous. If machine learning researchers are the target audience, many explanations of terms are unnecessary and remain redundant, since they are repeated in the text. If it is aimed at a broad general public, many of the methods mentioned are not explained enough and often seem to be strung together and poorly embedded in a context. This makes reading difficult to understand and not profitable. The ductus of writing here should be more explanatory and classifying, less enumerating. Examples would be helpful and illustrative. The somewhat verbose introduction could be shortened here in favor of more detailed explanations in the main part. Basically, the concrete and pragmatic writing style is very good, but in some cases illustrations or examples would be helpful.
2) Some readers who are not geneticists or deep learning experts may wish to have some help for everyday life on how to systematically evaluate variants. Here, a diagram with an exemplary workflow would be helpful.
3) The illustrations are basically rather weak and seem "cartoon-like" (fig. 3) or too "technical" (fig. 1). This is a pity, because here would be a great chance to generate much-noticed overview illustrations, which are gladly quoted.
Minor points:
1) line 65: Citation for frequency of GGEs is missing. 15-20%?
2) Line 73 + 99: Citation not found.
Author Response
Response to Reviewer 1 Comments
- Summary
We thank the reviewer for taking the time to review our review manuscript and for provinig comprehensive and helpful feedback. Please find below our responses and revisions have been made in track changes in there-submitted file.
Major points:
- Throughout, the target audience remains somewhat unclear. If this overview is intended for geneticists, a glossary is completely superfluous. If machine learning researchers are the target audience, many explanations of terms are unnecessary and remain redundant, since they are repeated in the text. If it is aimed at a broad general public, many of the methods mentioned are not explained enough and often seem to be strung together and poorly embedded in a context. This makes reading difficult to understand and not profitable. The ductus of writing here should be more explanatory and classifying, less enumerating. Examples would be helpful and illustrative. The somewhat verbose introduction could be shortened here in favor of more detailed explanations in the main part. Basically, the concrete and pragmatic writing style is very good, but in some cases illustrations or examples would be helpful.
Response: We appreciate the the concerns raised by the reviewer. To improve readability we have simplified the text in the introduction as suggested. To help contextualise the DL applications described in the text, we have replaced Fig. 1 with a figure that maps them to 3 main types of tasks required for genetic research reflected in the sub-headings in the text.
In relation to the target audience, we consider that epilepsy genetic research often requires a multidisciplinary approach and increasingly involves the integration of data and information from different research modalities and fields. We therefore envisage that the review will be of interest to individuals with diverse backgrounds and objectives. For example, a computational expert may be interested in applying methods they developed in another domain to the field of epilepsy, while an epilepsy clinician researcher might want to gain a basic understanding of new technologies that are influencing the field. In these scenarios, we consider the the glossaries could be useful or easily ignored as required.
Some readers who are not geneticists or deep learning experts may wish to have some help for everyday life on how to systematically evaluate variants. Here, a diagram with an exemplary workflow would be helpful.
Response: We agree that a reader may be unfamiliar with the processes involved in the evaluation and interpretation of genetic variants. We have now added a Figure that outlines this workflow.
- The illustrations are basically rather weak and seem "cartoon-like" (fig. 3) or too "technical" (fig. 1). This is a pity, because here would be a great chance to generate much-noticed overview illustrations, which are gladly quoted.
Thank you for this feedback. To address this we have removed Figure 3 and simplified Figure 1.
Minor points:
1) Line 65: Citation for frequency of GGEs is missing. 15-20%?
Response: Appropriate citations have now been inserted.
2) Line 73 + 99: Citation not found.
Response: These have been removed.
Reviewer 2 Report
At the manuscript “Applications for Deep Learning in Epilepsy Genetic Research” by Drs. Robert Zeibich et al. authors reviewed deep learning tools and discussed how they may be applied to the analysis of genetic data for epileptic studies. The authors have done a great job, the manuscript is very impressive. Some questions, from my point of view, require clarification.
The Arc and Homer1 genes have been identified as candidate genes involved in the pathogenesis of epilepsy and a number of comorbid diseases. In particular, it is likely that genes Arc and Homer1 may contribute to the coexistence of epilepsy and depression. It seems to me that these recent studies should be reflected in the presented manuscript. I would suggest using the following publications:
Borges et al, Intermittency properties in a temporal lobe epilepsy model; Epilepsy Behav. 2023 Feb;139:109072. doi: 10.1016/j.yebeh.2022.109072. Epub 2023 Jan 16. PMID: 36652897
The Arc and Homer1 genes have been identified as candidate genes involved in the pathogenesis of epilepsy and a number of comorbid diseases. In particular, it is likely that genes Arc and Homer1 may contribute to the coexistence of epilepsy and depression. It seems to me that these recent studies should be reflected in the presented manuscript. I would suggest using the following publications:
Haug et al, Mutation screening of the chromosome 8q24.3-human activity-regulated cytoskeleton-associated gene (ARC) in idiopathic generalized epilepsy; Mol Cell Probes. 2000 Aug;14(4):255-60. doi: 10.1006/mcpr.2000.0314.
Sibarov et al; Arc protein, a remnant of ancient retrovirus, forms virus-like particles, which are abundantly generated by neurons during epileptic seizures, and affects epileptic susceptibility in rodent models; Front. Neurol., 2023 ; Sec. Epilepsy; V 14 https://doi.org/10.3389/fneur.2023.1201104
Chen et al, Genetic and epigenetic mechanisms of epilepsy: a review Neuropsychiatr Dis Treat . 2017 Jul 13;13:1841-1859. doi: 10.2147/NDT.S142032. eCollection 2017.
Affecting the classification of forms of epilepsy, the authors do not mention the absence epilepsy. Perhaps this should have been mentioned at least briefly. Absence-epilepsy is a form of epilepsy in which seizures begin in childhood and most patients experience spontaneous remission during adolescence. The question of the genetic base of Absence-epilepsy remains unclear but in patients with absence-epilepsy some mutations have been found (for example, the GABRG2 gene, which encodes the GABA receptor subunit, and CACNA1A, which encodes the Ca 2+ channel subtype). Perhaps in such an extensive review it would make sense to add some material on this issue.
The presentation of a subject is systematic and comprehensive and analysis is proper. I am happy to recommend the manuscript for the publication after minor corrections mentioned above.
Round 2
Reviewer 1 Report
The authors addressed all of my comments. I can recommend this manuscript for acceptance in the present form.